# A comparison of penalised regression methods for informing the selection of predictive markers

Christopher J. Greenwood[1,2]*, George J. Youssef[1,2], Primrose Letcher[3], Jacqui A. Macdonald[1,2], Lauryn J. Hagg[1], Ann Sanson[3], Jenn Mcintosh[1,2], Delyse M. Hutchinson[1,2,3,4], John W. Toumbourou[1], Matthew Fuller-Tyszkiewicz[1], Craig A. Olsson[1,2,3]

1 Faculty of Health, School of Psychology, Centre for Social and Early Emotional Development, Deakin University, Geelong, Australia, 2 Centre for Adolescent Health, Murdoch Children's Research Institute, Melbourne, Australia, 3 Department of Paediatrics, Royal Children's Hospital, University of Melbourne, Melbourne, Australia, 4 Faculty of Medicine, National Drug and Alcohol Research Centre, University of New South Wales, Randwick, Australia

* christopher.greenwood@deakin.edu.au

**Data Availability Statement:** Ethics approvals for this study do not permit these potentially re-identifiable participant data to be made publicly available. Enquires about collaboration are possible

## Abstract

### Background

Penalised regression methods are a useful atheoretical approach for both developing predictive models and selecting key indicators within an often substantially larger pool of available indicators. In comparison to traditional methods, penalised regression models improve prediction in new data by shrinking the size of coefficients and retaining those with coefficients greater than zero. However, the performance and selection of indicators depends on the specific algorithm implemented. The purpose of this study was to examine the predictive performance and feature (i.e., indicator) selection capability of common penalised logistic regression methods (LASSO, adaptive LASSO, and elastic-net), compared with traditional logistic regression and forward selection methods.

### Design

Data were drawn from the Australian Temperament Project, a multigenerational longitudinal study established in 1983. The analytic sample consisted of 1,292 (707 women) participants. A total of 102 adolescent psychosocial and contextual indicators were available to predict young adult daily smoking.

### Findings

Penalised logistic regression methods showed small improvements in predictive performance over logistic regression and forward selection. However, no single penalised logistic regression model outperformed the others. Elastic-net models selected more indicators than either LASSO or adaptive LASSO. Additionally, more regularised models included fewer indicators, yet had comparable predictive performance. Forward selection methods

through our institutional data access protocol: https://lifecourse.melbournechildrens.com/data-access/. The current institutional body responsible for ethical approval is The Royal Children's Hospital Human Research Ethics Committee.

**Funding:** Data collection for the ATP study was supported primarily through Australian grants from the Melbourne Royal Children's Hospital Research Foundation, National Health and Medical Research Council, Australian Research Council, and the Australian Institute of Family Studies. Funding for this work was supported by grants from the Australian Research Council [DP130101459; DP160103160; DP180102447] and the National Health and Medical Research Council of Australia [APP1082406]. Olsson, C.A. was supported by a National Health and Medical Research Council fellowship (Investigator grant APP1175086). Hutchinson, D.M. was supported by the National Health and Medical Research Council of Australia [APP1197488].

**Competing interests:** The authors have declared that no competing interests exist.

dismissed many indicators identified as important in the penalised logistic regression models.

## Conclusions

Although overall predictive accuracy was only marginally better with penalised logistic regression methods, benefits were most clear in their capacity to select a manageable subset of indicators. Preference to competing penalised logistic regression methods may therefore be guided by feature selection capability, and thus interpretative considerations, rather than predictive performance alone.

## Introduction

Maximising prediction of health outcomes in populations is central to good public health practice and policy development. Population cohort studies have the potential to provide such evidence due to the collection of a wide range of developmentally appropriate psychosocial and contextual information on individuals over extended periods of time. It is however, challenging to identify which indicators maximise prediction, particularly when a large number of potential indicators is available. Atheoretical predictive modelling approaches–such as penalised regression methods–have the potential to identify key predictive markers while handling potential multicollinearity issues, addressing selection biases that impinge on the ability of indicators to be identified as important, and using procedures that enhance likelihood of replication. The interpretation of penalised regression is relatively straightforward for those accustomed to regressions, and thus represents an accessible solution. Furthermore, growing accessibility of predictive modelling tools is encouraging and presents as a potential point of advancement for identifying key predictive indicators relevant to population health [1].

Broadly, predictive modelling contrasts with the causal perspective most commonly seen within cohort studies, in that it aims to maximise prediction of an outcome, not investigate underlying causal mechanisms [2]. While both predictive and causal perspectives share some similarities, Yarkoni and Westfall note that "*it is simply not true that the model that most closely approximates the data-generating process will in general be the most successful at predicting real-world outcomes*" [2, p. 1000]. This reflects the perennial difficulty of achieving full or even adequate representation of underlying constructs through measurable data, creating a "*disparity between the ability to explain phenomena at the conceptual level and the ability to generate predictions at the measurable level*" [3, p. 293]. The perspectives differ further in their foci: while causal inference approaches are commonly interested in a single exposure-outcome pathway [4, 5], predictive modelling focuses on multivariable patterns of indicators that together predict an outcome [6].

Two of the key goals in finding evidence for predictive markers are improving the accuracy and generalisability of predictive models. Accuracy refers to the ability of the model to correctly predict an outcome, whereas generalisability refers to the ability of the model to predict well given new data [7]. The concept of *model fitting* is key to understanding both accuracy and generalisability. Over-fitting is of particular concern and refers to the tendency for analyses to mistakenly fit models to sample-specific random variation in the data, rather than true underlying relationships between variables [8]. When a model is overfitted, it is likely to be accurate in the dataset it was developed with but is unlikely to generalise well to new data.

Several model building considerations are key to balancing the accuracy and generalisability of predictive models. First and foremost is the use of *training* and *testing* data. Training

data are the data used to generate the predictive model, and testing data are then used to examine the performance of the predictive model. However, given the rarity of entirely separate cohort datasets suitable to train and then test a predictive model, a single data set is often split into training and testing portions; two subsets of a larger data pool [7]. If predictive models are trained and tested on the exact same data (i.e., no data splitting), the accuracy of models is likely to be inflated due to overfitting. To further improve generalisability, it is also recommended to iterate through a series of many training and testing data splits, to reduce the influence of any specific training/testing split of the data [9].

Another consideration important to balancing the accuracy and generalisability of predictive models is the process of *regularisation* involved in penalised regression. Regularisation is an automated method whereby the strength of coefficients for predictive variables that are deemed unimportant in predicting the outcome are shrunk towards zero. Regularisation also helps reduce overfitting by balancing the bias-variance trade-off [10]. Specifically, by reducing the size of the estimated coefficients (i.e., adding bias and reducing accuracy in the training data), the model becomes less sensitive to the characteristics of the training data, resulting in smaller changes in predictions when estimating the same model in the testing data (i.e., reducing variation and increasing generalisability).

Additionally, regularisation aids in balancing the accuracy and generalisability of predictive models in terms of *complexity*. Complexity often refers to the number of indicators in the final model. For several penalised regression procedures, the regularisation process results in the coefficients of unimportant variables being shrunk to zero (i.e., excluded from the model). Importantly, reducing the number of indicators in the final model helps to reduce overfitting. This is commonly referred to as feature selection [10]. Feature selection helps to improve the interpretability of models by selecting only the most important indicators from a potentially large initial pool, which is critical for researchers seeking to create administrable population health surveillance tools. While traditional approaches, such as backward elimination and forward selection procedures [11], are capable of identifying a subset of indicators, penalised regression methods improve on these by entering all potential variables simultaneously, reducing biases induced by the order variables are entered/removed from the model.

Retention of indicators is, however, influenced by the particular decision rules of the algorithm. Three penalised regression methods which conduct automatic feature commonly compared in the literature and discussed in standard statistical texts [10, 12] are the least absolute shrinkage and selection operator (LASSO) [13], the adaptive LASSO [14] and the elastic-net [15]. The LASSO applies the $L_1$ penalty (constraint based on the sum of the absolute value of regression coefficients), which shrinks coefficients equally and enables automatic feature selection. However, in situations with highly correlated indicators the LASSO tends to select one and ignore the others [16]. The adaptive LASSO and elastic-net are extensions on the LASSO, both of which incorporate the $L_2$ penalty from ridge regression [17].

The $L_2$ penalty (constraint based on the sum of the squared regression coefficients) shrinks coefficients equally towards zero, but not exactly zero, and is beneficial in situations of multicollinearity as correlated indicators tend to group towards each other [16, 18]. More specifically, the adaptive LASSO incorporates an additional data-dependent weight (derived from ridge regression) to the $L_1$ penalty term, which results in coefficients of strong indicators being shrunk less than the coefficients of weak indicators [14], contrasting to the standard LASSO approach. The elastic-net includes both the $L_1$ and $L_2$ penalty and enjoys the benefits of both automatic feature selection and the grouping of correlated predictors [15].

In addition to selecting between alternative penalised regression methods, analysts need to *tune* (i.e., select) the parameter lambda ($\lambda$), which controls the strength of the penalty terms. This is most commonly done via the data driven process of k-fold cross-validation [19].

Specifically, k-fold cross-validation splits the training data into k number of partitions, for which a model is built on k-1 of the partitions and validated on the remaining partition. This is repeated k number of times until each partition is used once as the validation data. 5-fold cross is often used. Commonly, the value of λ which minimises out-of-sample prediction error is selected, identifying the *best* model. An alternative parameterisation applies the *one-standard-error* rule, for which a value of λ is selected that results in the most regularised model which is within one standard error of the *best* model [20]. Comparatively, the *best* model usually selects a greater number of indicators than the *one-standard-error* model. The tuning of λ is, however, often poorly articulated and represents an important consideration for those seeking to derive a succinct set of predictive indicators [21].

The purpose of this study was to examine the predictive performance and feature (i.e., variable) selection capability of common penalised logistic regression methods (LASSO, adaptive LASSO, and elastic-net) compared with traditional logistic regression and forward selection methods. To demonstrate methods, a broad range of adolescent health and development indicators, drawn from one of Australia's longest-running cohort studies of social-emotional development, was used to maximise prediction of tobacco use in young adulthood. Tobacco use is widely recognised as a leading health concern in Australia [22], making it a high priority area for government investment [23] and a targetable health outcome of predictive models. Although some comparative work has previously been examined the prediction substance use with cohort study data [24], comparisons have largely focused on differences in predictive performance, not feature selection, and have yet to examine differences between the *best* and *one-standard-error* models.

## Method

### Participants

Participants were from the Australian Temperament Project (ATP), a large multi-wave longitudinal study (16 waves) tracking the psychosocial development of young people from infancy to adulthood. The baseline sample in 1983 consisted of 2,443 infants aged between 4–8 months from urban and rural areas of Victoria, Australia. Information regarding sample characteristics and attrition are available elsewhere [25, 26]. The current sample consisted of 1292 (707 women) participants with responses from at least one of the adolescent data collection waves (ages 13–14, 15–16 or 17–18 years) and who remained active in the study during at least one of the three young adult waves (ages 19–20, 23–24 or 27–28 years).

Research protocols were approved by the Human Research Ethics Committee at the University of Melbourne, the Australian Institute of Family Studies and/or the Royal Children's Hospital, Melbourne. Participants' parents or guardians provided informed written consent at recruitment into the study, and participants provided informed written consent at subsequent waves.

### Measures

**Adolescent indicators.** A total of 102 adolescent indicators, assessed at ages 13–14, 15–16, 17–18 years, by parent and self-report, were available for analysis (see S1 Table). Data spanned individual (i.e., biological, internalising/externalising, personality/temperament, social competence, and positive development), relational (i.e., peer and family relationships, and parenting practices), contextual (demographics, and school and work), and substance use specific (personal, and environmental use) domains. Repeated measures data were combined (i.e., maximum or mean level depending on the indicator type) to represent overall adolescent experience. All 102 adolescent indicators were entered as predictors into each model.

**Young adult tobacco use outcome.** Tobacco use was assessed at age 19–20 years (after measurement of the indicators), as the number of days used in the last month. This was converted to a binary variable representing daily use (i.e., $\geq 28$ days in the last month), which was used as the outcome (response variable) in all analyses.

## Statistical analyses

R statistical software [27] was used for all analyses. Since standard penalised regression packages in R do not have in-built capacity to handle multiply imputed or missing data, a single imputed data set was generated using the *mice* package [28]. Following imputation, continuous indicators were standardised by dividing scores by two times its standard deviation, to improve comparability between continuous and binary indicators [29].

**Regression procedures.** Logistic, LASSO, adaptive LASSO, and elastic-net logistic regressions were run using the *glmnet* package [16]. The additional weight term used in the adaptive LASSO was derived as the inverse of the corresponding coefficient from ridge regression. Elastic-net models were specified as using an equal split between $L_1$ and $L_2$ penalties. Specifically, $L_1$ penalization imposes a constraint based on the sum of the absolute value of regression coefficients, whilst $L_2$ penalisation, imposes a constraint based on the sum of the squared regression coefficients [12]. 5-fold cross-validation was used to tune $\lambda$ (the strength of the penalty) for all penalised logistic regression methods. Logistic regression imposes no penalisation on regression coefficients. Forward selection logistic regression was conducted using the *MASS* package [30]. Predictive performance and feature selection were examined separately using the processes described below. Models were run based on an adapted procedure and syntax implemented by Ahn et al. [9]. All syntax is available online (https://osf.io/ehprb/?view_only= 9f96d224f08e4987829bb29204061f4b).

The process to compare predictive performance is illustrated in Fig 1A. All models were implemented in 100 iterations of training and testing data splits (80/20%). For penalised logistic regression approaches, 100 iterations of 5-fold cross-validation were implemented to tune $\lambda$ in the training data (80%). Each iteration of cross-validation identified the *best* ($\lambda$-min; model which minimised out-of-sample predictor error) and *one-standard-error* ($\lambda$-1se; more regularised model with out-of-sample prediction error within one standard error of the *best* model) model. For all models, predictions were made in the test data (20%). For penalised logistic regression approaches, mean predictive performance was derived across cross-validation iterations. For all models, predictive performance metrics were saved for each of the 100 training and testing data splits.

Predictive performance was assessed using the area under the curve (AUC) of the receiver operator characteristic [31] and the harmonic mean of precision and recall (F1 score) [32]. AUC indicates a model's ability to discriminate a given binary outcome, by plotting the true positive rate (the likelihood of correctly identifying a case) against the false positive rate (i.e., the likelihood of incorrectly identifying a case). An AUC value of 0.5 is equivalent to chance, and a value of 1 equals perfect discrimination [33]. However, as base rates decline (i.e., the prevalence of the outcome gets lower), the AUC can become less reliable because high scores can be driven by correctly identifying true-negatives, rather than true-positives. When base rates are low, the F1 score is a useful addition [34]. The F1 score represents the harmonic average of precision (i.e., proportion of true-positives from the total number of positives identified) and recall/sensitivity (i.e., the proportion of true-positive identified from the total number of true-positives). The F1 score indicates perfect prediction at 1 and inaccuracy at 0.

The process to compare feature selection capability is illustrated in Fig 1B. In line with Ahn et al [9], to identify the most robust indicators from each model, similar procedures to that

## A. Predictive performance

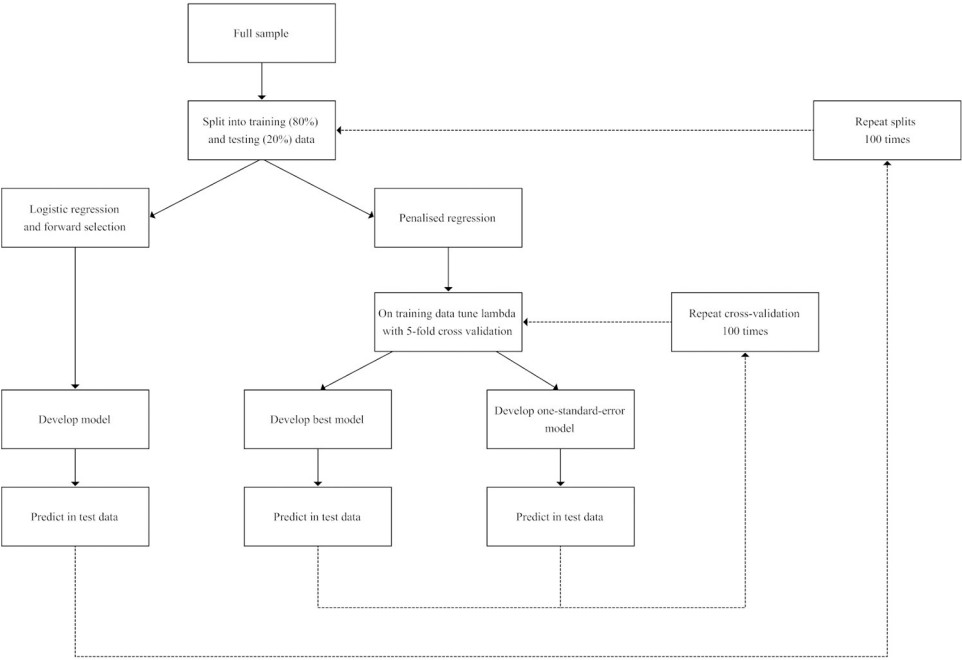

## B. Feature selection

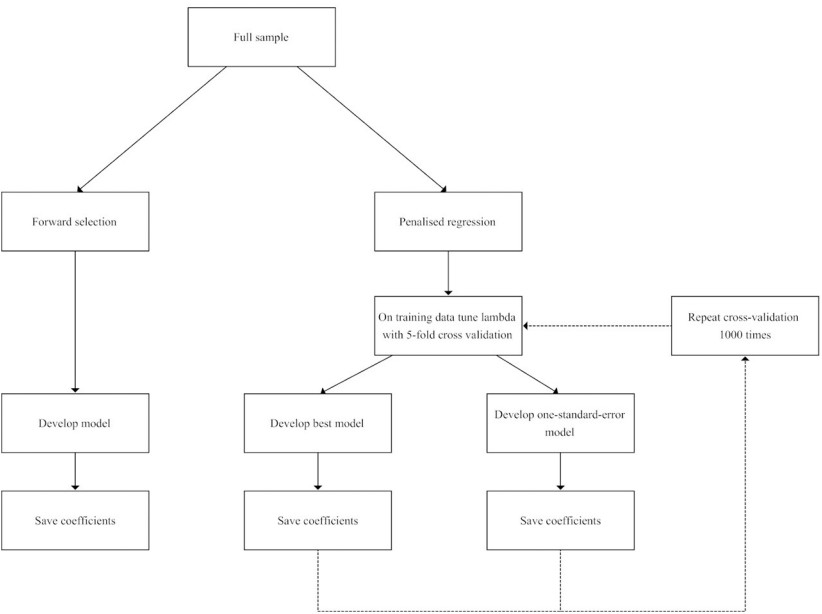

**Fig 1.** A. Process to compare predictive performance of models. B. Process to compare feature selection capability of models.

described above were implemented; although the full data set was used to train the models. Additionally, the number of cross-validation iterations was increased to 1,000. For the penalised logistic regression methods, robust indicators were considered as those which were

selected in at least 80% of the cross-validation iterations. For indicators that met this robust criterion the mean of the coefficients was taken, whilst others were set to zero. Small coefficients were considered as those between -0.1 and 0.1 [34].

## Results

### Predictive performance

The predictive performance of each model across iterations of training and testing data splits is presented in Fig 2.

All λ-min penalised logistic regression models had higher AUC scores than both logistic regression and forward selection (Δ median AUC 0.002–0.008). Similarly, all of the λ-1se penalised logistic regression models outperformed forward selection (Δ median AUC 0.003–0.006), however, only the λ-1se elastic-net model outperformed logistic regression (Δ median AUC 0.002). Similarly, in comparison to logistic regression and forward selection F1 scores were higher for all λ-min and λ-1se penalised logistic regression models (Δ median F1 0.007–0.025).

Between penalised logistic regression models, the elastic-net had the highest AUC scores within both the λ-min (Δ median AUC 0.001–0.002) and λ-1se (Δ median AUC 0.003) models.

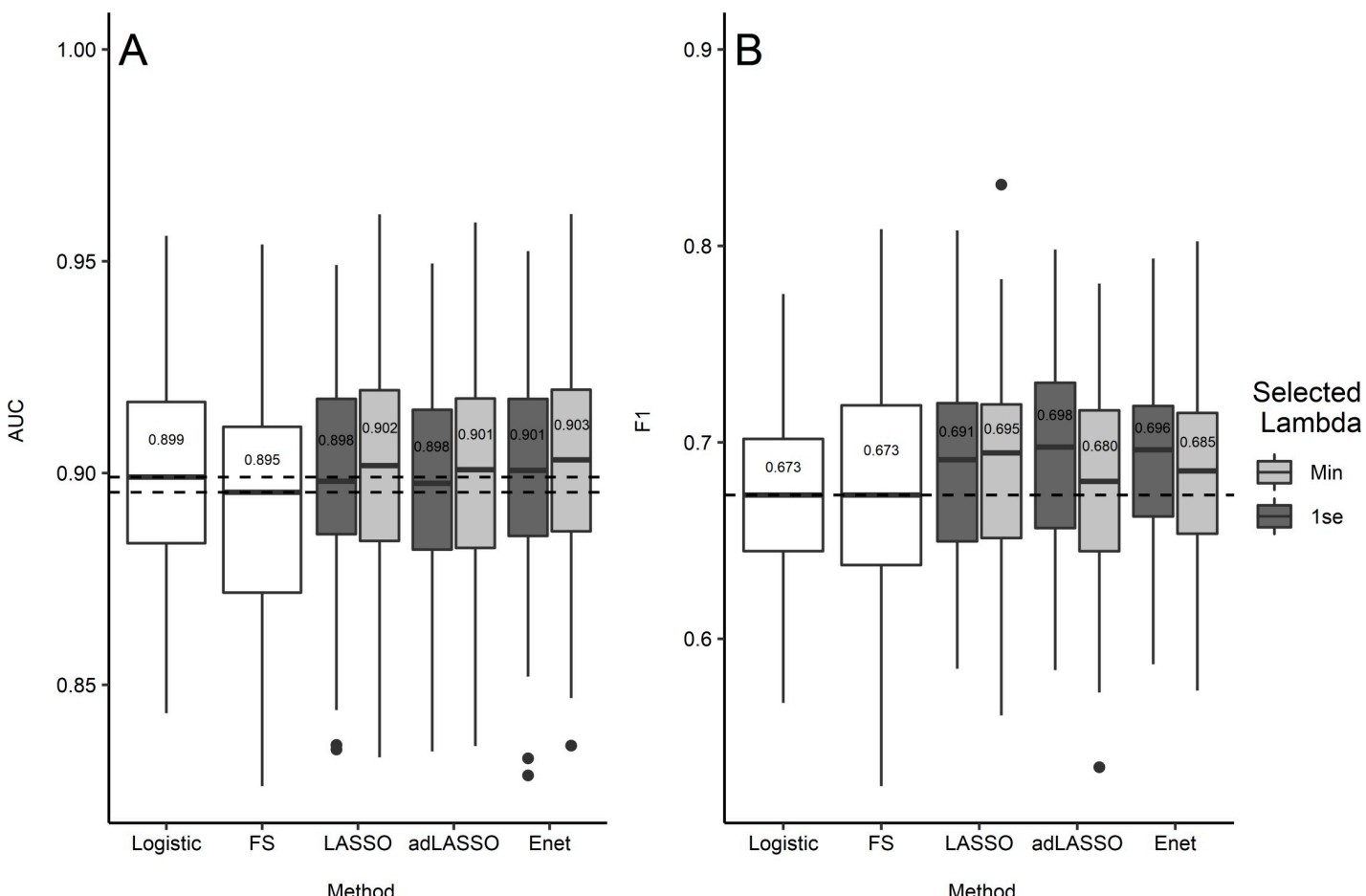

**Fig 2.** Predictive performance: Box and whisker plot of AUC (A) and F1 (B) scores across 100 iterations of training and testing data splits. Dotted lines indicating median performance for logistic regression and forward selection; Median values reported for each model.

Comparatively, the LASSO had the highest F1 scores for the λ-min models (Δ median F1 0.010–0.015) and the adaptive LASSO scored highest F1 scores for the λ-1se models (Δ median F1 0.002–0.007).

Finally, all λ-min models had higher AUC scores than the respective λ-1se models (Δ median AUC 0.002–0.004). In contrast, while the λ-min LASSO had the higher F1 score than the λ-1se model (Δ median F1 0.004), the λ-1se model outperformed the respective λ-min model for adaptive LASSO and elastic-net (Δ median F1 0.011–0.018).

**Feature selection.** Feature selection was compared between penalised logistic regression methods and forward selection logistic regression. Fig 3 plots the beta coefficients from feature selection methods. Indicators were only presented if selected in at least one model.

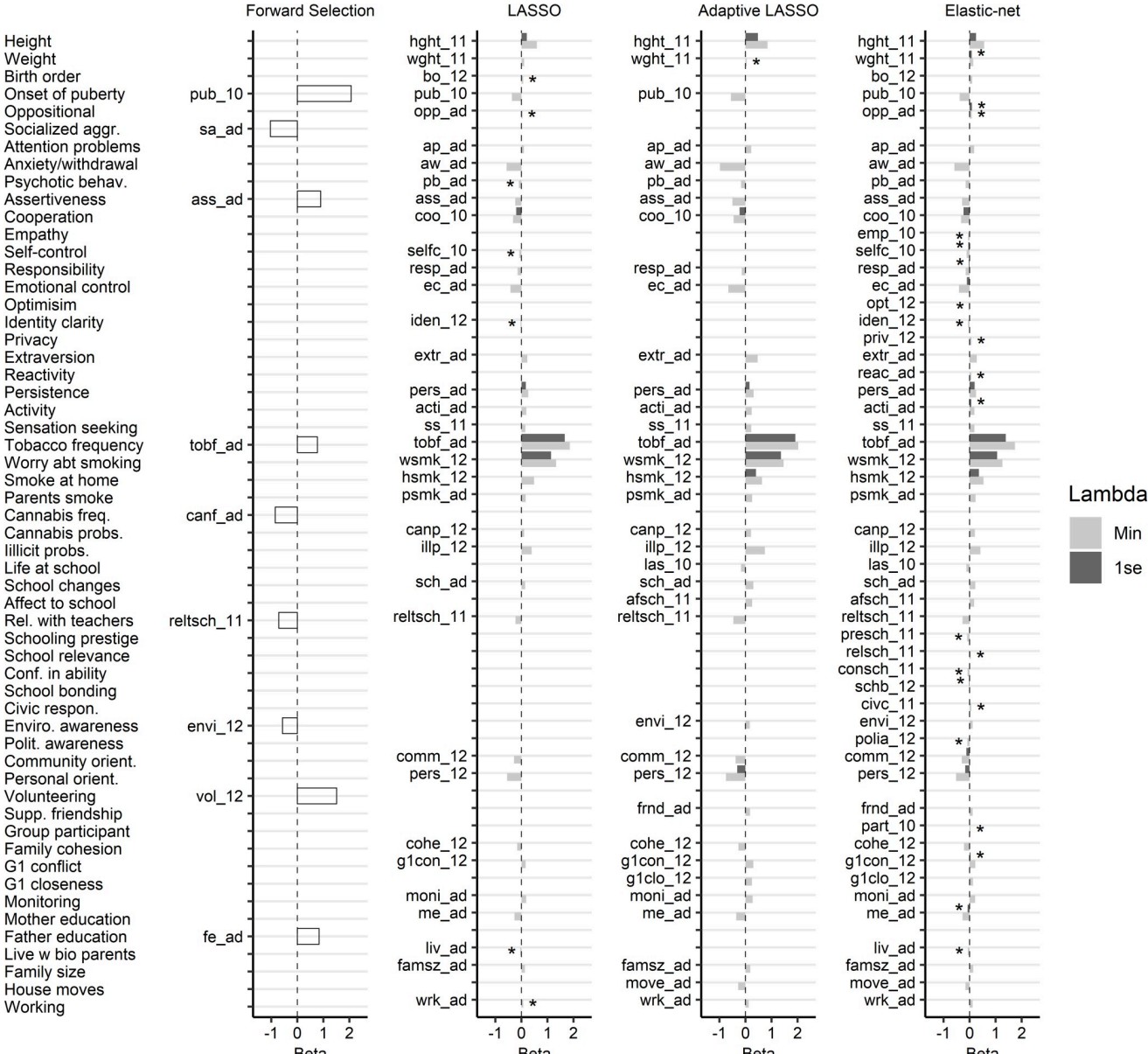

**Fig 3. Mean beta coefficients for indicators that survived at least 80% of 1,000 iterations;** * = small coefficients (between -0.1 and 0.1). Suffix 10 = wave 10 (13–14 years) only; 11 = wave 11 (15–16 years) only; 12 = wave 12 (17–18 years) only; ad = combined across adolescence.

There were notable differences in feature selection when comparing the penalised logistic regression methods to the forward selection procedure. Specifically, the number of indicators selected via forward selection was far below that selected in the λ-min models. There was, however, similarity in the number of selected indicators between forward selection and the λ-1se models. Even so, forward selection models selected several indicators not present in any of the penalised logistic regression models.

There was notable similarity in the number of indicators selected in the LASSO and adaptive LASSO. The LASSO did, however, select slightly fewer indicators than the adaptive LASSO for both the λ-min (35 v 36) and λ-1se (5 v 7) models. Comparatively, both the LASSO and adaptive LASSO λ-min models contained several unique indicators (five and six, respectively); in the LASSO unique indicators were limited to those with small coefficients, whereas in the adaptative LASSO none of the unique indicators had small coefficients. Additionally, while the λ-min LASSO contained a total of seven indicators with small coefficients (28 indicators with non-small coefficients), the respective adaptive LASSO only contained one (35 indicators with non-small coefficients). There was clear similarity between the LASSO and adaptive LASSO λ-1se models, such that the adaptive LASSO contained all indicators from the LASSO (and two unique indicators). For both the LASSO and adaptive LASSO λ-1se models, no indicators had small coefficients.

In contrast, the elastic-net models selected more indicators in both the λ-min (51) and λ-1se models (17) than the respective LASSO and adaptive LASSO models. The elastic-net models selected all indicators selected with the LASSO and adaptive LASSO. Additionally, there were several unique elastic-net λ-min and λ-1se indicators (10 and 10, respectively); almost all of these unique indicators had small coefficients. The elastic-net selected a total of 13 indicators with small-coefficients from the λ-min model (38 indicators with non-small coefficients) and 8 from the λ-1se model (9 indicators with non-small coefficients).

## Discussion

This study examined three common penalised logistic regression methods, LASSO, adaptive LASSO and elastic-net, in terms of both predictive performance and feature selection using adolescent data from a mature longitudinal study of social and emotional development to maximise prediction of daily tobacco use in young adulthood. We demonstrated an analytical process for examining predictive performance and feature selection and found that while differences in predictive performance were only small, differences in feature selection were notable. Findings suggested that the benefits of penalised logistic regression methods were most clear in their capacity to select a manageable subset of indicators. Therefore, decisions to select any particular method may benefit from reflecting on interpretative considerations.

The use of penalised regression approaches provides one method of identifying important indicators from amongst a large pool. While the interpretation of penalised regression methodologies is relatively straight forward for those accustomed to regression analyses, the process by which output is obtained requires a series of iterative procedures, which may be somewhat novel. This study has outlined one potential approach to this sequence of analyses and provided syntax for others to apply and adjust analyses for themselves. Understanding both the predictive performance and the feature selection capabilities of any one model is necessary to make informed decisions regarding the development of population surveillance tools. For instance, a well predicting model with too many indicators may be difficult to translate into a succinct tool, whereas an interpretable model with poor predictive performance may throw into question the usefulness of such indicators. Additionally, the current analytical process

allows for the examination of both the *best* and, more regularised, *one-standard-error* models, which have previously received limited attention.

Findings suggested greater predictive performance of penalised logistic regression compared to logistic regression or forward selection, albeit small. Results were consistent in that all of the *best* models outperformed both standard and forward selection logistic regression in terms of both AUC and F1 scores. Additionally, all *one-standard-error* models outperformed forward selection on both performance indices. The only underperforming models were the *one-standard-error* LASSO and adaptive LASSO, which had lower AUC scores, albeit only marginally, than standard logistic regression. The standard logistic regression models, however, included no feature selection and thus is not a useful alternative.

Current findings are supportive of previously identified improvements in predictive performance, although improvements appear to be smaller than previously documented [24]. While, alignment with previous work may be limited by the substantial differences in both indicators and outcomes, even small improvements in predictive performance encourage the use of penalised logistic regression methods. In contrast to previous literature [24], differences in predictive performance between penalised logistic regression methods were only small and did not suggest any particular best performing method. Specifically, elastic-net models did not show an improvement in predictive performance over other methods, whereby, although elastic-net models had the highest AUC scores, the highest F1 scores were found in the LASSO and adaptive LASSO models. For those seeking to use penalised logistic regression methods to develop population surveillance tools, the current findings suggest that all examined methods performed similarly.

There were, however, notable differences in feature selection–and thus interpretation– between models. Forward selection, in comparison to the penalised logistic regression approaches, selected far fewer indicators than the *best* penalised logistic regression models. Additionally, forward selection included a number of indicators not selected in the penalised logistic regression models and omitted indicators with large coefficients identified in the penalised logistic regression models. Overall, there appeared little benefit to the use of forward selection over penalised logistic regression alternatives. This recommendation coincides with a range of statistical concerns inherent to forward selection procedures [35].

In comparing among penalised logistic regression models, most notably, elastic-net models selected substantially more indicators than either the LASSO or adaptive LASSO, although almost all indicators unique to elastic-net models had small coefficients. The elastic-net selected all indicators included in the penalised logistic regression models. Additionally, while the LASSO and adaptive LASSO selected a similar number of indicators, both models contained several unique indicators. All indicators unique to the LASSO, however, had small effects. whereas in the adaptive LASSO unique indicators had more pronounced effects. These findings suggest a considerable level of similarity in the selected features, with differences between models largely limited to indicators with small coefficients.

A particularly relevant consideration for those seeking to develop predictive population surveillance tools is the differences in feature selection between the *best* and *one-standard-error* models, which reflects understanding the desired goal of the model [1]. Findings suggest that the *one-standard-error* models selected far fewer indicators than the *best* models yet had relatively comparable predictive performance. In developing a predictive population surveillance tool which is intended to be relevant to a diversity of outcomes (e.g., tobacco use or mental health problems), examining both the *best* and *one-standard-error* models simultaneously is likely to convey advantage in determining which indicators are worth retaining. By examining the *best* model, a diverse range of indicators are likely to be identified, for which indicators (even those with weak prediction) may share overlap across multiple outcomes and suggest

cost effective points of investment. By examining the *one-standard-error* model, the smallest subset of predictors for each relevant outcome can be identified, which may suggest the most pertinent indicators for future harms.

There are, however, some study limitations to note. First, the use of real data provides a relatable demonstration of how models may function, but findings may not necessarily generalise to other populations or data types. The use of simulation studies remains an important and complementary area of research for systematically exploring differences in predictive models [e.g., 12]. Second, we did not explore all available penalised logistic regression applications, but rather methods that were common and accessible to researchers. Methods such as the relaxed LASSO [36] or data driven procedures to balance the $L_1$ and $L_2$ penalties of elastic-net [34] require similar comparative work. Finally, as penalised regression methods have not been widely implemented into a MI framework, the current study relies on a singly-imputed data set [37, 38].

In summary, this paper provided an overview of the implementation and both the predictive performance and feature selection capacity of several common penalised logistic regression methods and potential parameterisations. Such approaches provide an empirical basis for selecting indicators for population surveillance research aimed at maximising prediction of population health outcomes over time. Broadly, findings suggested that penalised logistic regression methods showed improvements in predictive performance over logistic regression and forward selection, albeit small. However, in selecting between penalised logistic regression methods, there was no clear best predicting method. Differences in feature selection were more apparent, suggesting that interpretative goals may be a key consideration for researchers when making decisions between penalised logistic regression methods. This includes greater consideration of the respective *best* and *one-standard-error* models. Future work should continue to compare the predictive performance and feature selection capacities of penalised logistic regression models.

## Supporting information

**S1 Table. Description of adolescent indicators.** Note: a = approach to combine repeated measures data; SR = Self report, PR = Parent report; SMFQ = Short Mood and Feelings Questionnaire [39], RBPC = Revised Behaviour Problem Checklist [40], RCMAS = Revised Children's Manifest Anxiety Scale [41], SRED = Self-Report Early Delinquency Instrument [42], SSRS = Social Skills Rating System [43], CSEI = Coopersmith Self-Esteem Inventory [44], PIES = Psychosocial Inventory of Ego Strengths [45], ACER SLQ = ACER School Life Questionnaire [46], OSBS = O'Donnell School Bonding Scale [47], IPPA = Inventory of Parent and Peer Attachment [48], CBQ = Conflict Behaviour Questionnaire [49], FACES II = Family Adaptability and Cohesion Evaluation Scale [50], RAS = Relationship Assessment Scale [51], OHS = Overt Hostility Scale [52], ZTAS = Zuckerman's Thrill and Adventure Seeking Scale [53], FFPQ = Five Factor Personality Questionnaire [54], GBFM = Goldberg's Big Five Markers [55], SATI = School Age Temperament Inventory [56], more information on ATP derived scales can be found in Vassallo and Sanson [25].
(DOCX)

## Acknowledgments

The ATP study is located at The Royal Children's Hospital Melbourne and is a collaboration between Deakin University, The University of Melbourne, the Australian Institute of Family Studies, The University of New South Wales, The University of Otago (New Zealand), and the

Royal Children's Hospital (further information available at www.aifs.gov.au/atp). The views expressed in this paper are those of the authors and may not reflect those of their organizational affiliations, nor of other collaborating individuals or organizations. We acknowledge all collaborators who have contributed to the ATP, especially Professors Ann Sanson, Margot Prior, Frank Oberklaid, John Toumbourou and Ms Diana Smart. We would also like to sincerely thank the participating families for their time and invaluable contribution to the study.

## Author Contributions

**Conceptualization:** Christopher J. Greenwood, George J. Youssef, Craig A. Olsson.

**Data curation:** Christopher J. Greenwood, George J. Youssef.

**Formal analysis:** Christopher J. Greenwood, George J. Youssef.

**Funding acquisition:** Ann Sanson, Craig A. Olsson.

**Investigation:** Primrose Letcher, Jacqui A. Macdonald, Ann Sanson, Jenn Mcintosh, Delyse M. Hutchinson, John W. Toumbourou, Craig A. Olsson.

**Methodology:** Christopher J. Greenwood, George J. Youssef.

**Project administration:** Primrose Letcher, Craig A. Olsson.

**Software:** Christopher J. Greenwood, George J. Youssef.

**Supervision:** George J. Youssef.

**Visualization:** Christopher J. Greenwood.

**Writing – original draft:** Christopher J. Greenwood, George J. Youssef, Primrose Letcher, Jacqui A. Macdonald, Lauryn J. Hagg, Craig A. Olsson.

**Writing – review & editing:** Christopher J. Greenwood, George J. Youssef, Primrose Letcher, Jacqui A. Macdonald, Lauryn J. Hagg, Ann Sanson, Jenn Mcintosh, Delyse M. Hutchinson, John W. Toumbourou, Matthew Fuller-Tyszkiewicz, Craig A. Olsson.

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
