## [Decision Letter · Decision Letter 0]

3 Sep 2020

PONE-D-20-18792

An overview of penalised regression methods for informing the selection of predictive markers

PLOS ONE

Dear Dr. Christopher Greenwood,

Thank you for submitting your manuscript to PLOS ONE. After careful consideration, we feel that it has merit but does not fully meet PLOS ONE’s publication criteria as it currently stands. Therefore, we invite you to submit a revised version of the manuscript that addresses the points raised during the review process.

Please review the comments from the two reviewers and make appropriate corrections.

We look forward to receiving your revised manuscript.

Kind regards,

Yuka Kotozaki

Academic Editor

PLOS ONE

Journal Requirements:

Reviewers' comments:

Reviewer's Responses to Questions

**Comments to the Author**

1. Is the manuscript technically sound, and do the data support the conclusions?

Reviewer #1: Yes

Reviewer #2: No

2. Has the statistical analysis been performed appropriately and rigorously? 

Reviewer #1: Yes

Reviewer #2: No

3. Have the authors made all data underlying the findings in their manuscript fully available?

Reviewer #1: Yes

Reviewer #2: No

4. Is the manuscript presented in an intelligible fashion and written in standard English?

Reviewer #1: Yes

Reviewer #2: No

5. Review Comments to the Author

Reviewer #1: The manuscript is a useful, important, and largely well-written study on the applications of penalized logistic regression on real cohort data. I can envisions its eventual publication in PLOS ONE after suitable revisions are made.

Suggested Revisions

Abstract: It does not seem that penalized regression provides an "Atheoretical approach".

For example, the LASSO estimates of coefficients, under a Bayesian theoretical point of view, correspond to a posterior mode of the coefficients under independent Laplace priors.

Abstract: "Penalized regression methods" -> "Penalized logistic regression methods"

Throughout the manuscript:

Please provide page numbers for the manuscript.

Many references are made to "penalized regression" (e.g. LASSO, Elastic net) being used.

That phrase refers to real-valued responses. But the manuscript seems to focus on the use of logistic regression. So it is better to state "penalized logistic regression" instead of just "penalized regression" in these instances.

Lines 62-63: Can you give recent few examples of cohort studies that represent limited use of penalized regression in such studies?

Lines 82-83: "apply well to new data" is a vague term. Do you mean "predict new data well?"

Lines 191-192: "A single imputed data set was generated via multiple imputation (MI)" is an awkward phrase that needs to be fixed.

"Single imputation" is not an example of "multiple imputation".

Lines 193-194: The statement "since standard penalised regression procedures in R cannot handle multiple imputation or missing data"

s not exactly true. In principle, one can adopt a fully Bayesian approach to elastic net, lasso or ridge logistic regression, by running and MCMC algorithm generate samples from the posterior distribution of model parameters, for each imputed data set, from the multiple imputations of data sets. Then results are summarized by mixing together the MCMC posterior samples across data sets.

This is according to the Bayesian book: Bayesian Data Analysis (2nd Edition), by Andrew Gelman, et al.

Lines 197-238: The section on Regression Procedures lacks detailed descriptions of these procedures.

When describing each of the methods, namely the logistic, LASSO, adaptive LASSO, and elastic-net logistic regressions,

the section needs to describe the penalized likelihood function that is being maximized, by each method.

(Of course, the logistic regression method uses zero penalty).

For ideas, a good example of a paper that describes the likelihood function of each of these methods is provided in

the article described in https://www.jstatsoft.org/article/view/v033i01

Lines 212-213: Please define "best (lambda-min)" and "best one-standard-error (lambda-1se) model."

This will greatly help interpretation of the Results section, later.

The above two critical points was the main factor that moved my decision from Minor Revision to Major Revision.

Line 377: The phrase "single MI data set" a misnomer, because it refers to one thing having the property of being both "single" and "multiple". Instead, use something like "singly-imputed" data set.

Line 422: "BMJ". Spell out the journal's full name.

Line 426: "Int J Methods Psychiatr Res." Spell out the journal's full name.

Line 460: The reference is missing the volume, issue, and page numbers.

Line 478: The reference is missing the conference location.

Line 501: "BMJ". Spell out the journal's full name.

Line 532: The reference is missing the volume, issue, and page numbers.

Line 534: "Ssrn"appears to be incomplete journal name.

Line 539: The reference is missing the journal name.

Line 543: In the reference, capitalize the second "the".

Reviewer #2: As the title suggests that the manuscript presumably overviews the ability of different penalized regression methods in selecting predictive markers, it is not clear whether feature selection or predictability of selected features is focused. On the other hand, it is known that penalized regression methods are developed for variable selection instead of prediction, and there are a lot of theoretical results on the different methods (in contrast to the claimed "atheoretical predictive modelling approaches" by the authors). As long as the prediction is concerned, there are many machine learning methods (especially deep learning should be resorted). There are some other major issues:

1. The current overview solely relies on the analysis of single data set. Instead a comprehensive review may be better conducted with a well-designed simulation study, as well as several typical data sets. Otherwise, how have the authors calculated the sensitivity or specificity of selected predictors?

2. The presented analysis of the longitudinal study is very vague, lack of details. For example, what is the response variable, how many predictors are there, and what is the sample size? It suggested in "Discussion" on "prediction of daily tobacco use in young adulthood", does it mean a binary response variable, or quantitative response variable? As the original study suggests a longitudinal data, is a cross-sectional data set selected for the analysis? Otherwise, how is the longitudinal data fit to the proposed models (again lack of details)?

3. Logistic regression is a regression model, instead of a statistical approach in parallel of penalized regression method. As logistic regression model can be used here, I suppose a binary response variable is used here. In this case, have the authors considered all penalized regression methods for the logistic regression models?

6. PLOS authors have the option to publish the peer review history of their article (what does this mean?). If published, this will include your full peer review and any attached files.

Reviewer #1: No

Reviewer #2: No

---

## [Author Response · Author response to Decision Letter 0]

28 Oct 2020

October 22, 2020

Dr Yuka Kotozaki

RE: Revised manuscript for consideration in PLOS ONE

Dear Dr Yuka Kotozaki,

Thank you for the invitation to submit a revised version of our manuscript to PLOS ONE. We are grateful to the reviewers for their considered and helpful comments. We have responded to each and have revised the manuscript accordingly. We believe the manuscript has been strengthened through the peer review process. 

Yours sincerely,

Christopher Greenwood

Research Fellow | Life-course Epidemiology

Centre for Social and Early Emotional Development

Deakin University, Australia

E: christopher.greenwood@deakin.edu.au

Comments from the reviewers

Reviewer #1:

The manuscript is a useful, important, and largely well-written study on the applications of penalized logistic regression on real cohort data. I can envision its eventual publication in PLOS ONE after suitable revisions are made.

 Thank you for your kind feedback and suggestions for improving the manuscript.

Throughout the manuscript:

Please provide page numbers for the manuscript. 

 Done.

Many references are made to "penalized regression" (e.g. LASSO, Elastic net) being used.

That phrase refers to real-valued responses. But the manuscript seems to focus on the use of logistic regression. So it is better to state "penalized logistic regression" instead of just "penalized regression" in these instances.

 This has been amended throughout the revised manuscript so that we now consistently refer to penalized logistic regression instead of penalized regression.

Lines 62-63: Can you give recent few examples of cohort studies that represent limited use of penalized regression in such studies?

We have now re-written this part of the text to better express our point. The amended text now reads:

“Furthermore, growing accessibility of predictive modelling tools is encouraging and presents as a potential point of advancement for identifying key predictive indicators relevant to population health (Shatte et al., 2019).”

Lines 82-83: "apply well to new data" is a vague term. Do you mean "predict new data well?"

 Thank you, this has been clarified.

“Accuracy refers to the ability of the model to correctly predict an outcome, whereas generalisability refers to the ability of the model to predict well given new data (Altman et al., 2009).”

Lines 191-192: "A single imputed data set was generated via multiple imputation (MI)" is an awkward phrase that needs to be fixed. "Single imputation" is not an example of "multiple imputation".

Lines 193-194: The statement "since standard penalised regression procedures in R cannot handle multiple imputation or missing data" is not exactly true. In principle, one can adopt a fully Bayesian approach to elastic net, lasso or ridge logistic regression, by running and MCMC algorithm generate samples from the posterior distribution of model parameters, for each imputed data set, from the multiple imputations of data sets. Then results are summarized by mixing together the MCMC posterior samples across data sets. This is according to the Bayesian book: Bayesian Data Analysis (2nd Edition), by Andrew Gelman, et al.

 We thank the reviewer for this information. This section has now been clarified. 

“Since standard penalised regression packages in R do not have in-built capacity to handle multiply imputed or missing data, a single imputed data set was generated using the mice package (Van Buuren & Groothuis-Oudshoorn, 2011).”

Lines 197-238: The section on Regression Procedures lacks detailed descriptions of these procedures.

When describing each of the methods, namely the logistic, LASSO, adaptive LASSO, and elastic-net logistic regressions, the section needs to describe the penalized likelihood function that is being maximized, by each method. (Of course, the logistic regression method uses zero penalty).

For ideas, a good example of a paper that describes the likelihood function of each of these methods is provided in the article described in https://www.jstatsoft.org/article/view/v033i01

We have now described in the “Regression procedures” section the penalty terms.

“Specifically, L1 penalization imposes a constraint based on the sum of the absolute value of regression coefficients, whilst L2 penalisation, imposes a constraint based on the sum of the squared regression coefficients (Pavlou et al., 2016). 5-fold cross-validation was used to tune λ (the strength of the penalty) for all penalised logistic regression methods. Logistic regression imposes no penalisation on regression coefficients.”

We have additionally expanded the section of the introduction which discusses the compromises of each model and the relevant penalties. We have provided the full paragraph here for completeness:

“Retention of indicators is, however, influenced by the particular decision rules of the algorithm. Three penalised regression methods which conduct automatic feature commonly compared in the literature and discussed in standard statistical texts (James et al., 2013; Pavlou et al., 2016) are the least absolute shrinkage and selection operator (LASSO) (Tibshirani, 1996), the adaptive LASSO (Zou, 2006) and the elastic-net (Zou & Hastie, 2005). The LASSO applies the L1 penalty (constraint based on the sum of the absolute value of regression coefficients), which shrinks coefficients equally and enables automatic feature selection. However, in situations with highly correlated indicators the LASSO tends to select one and ignore the others (Friedman et al., 2010). The adaptive LASSO and elastic-net are extensions on the LASSO, both of which incorporate the L2 penalty from ridge regression (Cessie & Houwelingen, 1992). 

The L2 penalty (constraint based on the sum of the squared regression coefficients) shrinks coefficients equally towards zero, but not exactly zero, and is beneficial in situations of multicollinearity as correlated indicators tend to group towards each other (Feig, 1978; Friedman et al., 2010). More specifically, the adaptive LASSO incorporates an additional data-dependent weight (derived from ridge regression) to the L1 penalty term, which results in coefficients of strong indicators being shrunk less than the coefficients of weak indicators (Zou, 2006), contrasting to the standard LASSO approach. The elastic-net includes both the L1 and L2 penalty and enjoys the benefits of both automatic feature selection and the grouping of correlated predictors (Zou & Hastie, 2005).”

Lines 212-213: Please define "best (lambda-min)" and "best one-standard-error (lambda-1se) model."

This will greatly help interpretation of the Results section, later.

We have now added a description of these models to help with interpretation in the results section.

“Each iteration of cross-validation identified the best (λ-min; model which minimised out-of-sample predictor error) and one-standard-error (λ-1se; more regularised model with out-of-sample prediction error within one standard error of the best model) model.”

The above two critical points was the main factor that moved my decision from Minor Revision to Major Revision.

Line 377: The phrase "single MI data set" a misnomer, because it refers to one thing having the property of being both "single" and "multiple". Instead, use something like "singly-imputed" data set.

This has now been amended as “a singly-imputed data set”. 

Line 422: "BMJ". Spell out the journal's full name.

Done.

Line 426: "Int J Methods Psychiatr Res." Spell out the journal's full name.

Done.

Line 460: The reference is missing the volume, issue, and page numbers.

Done.

Line 478: The reference is missing the conference location.

Done.

Line 501: "BMJ". Spell out the journal's full name.

Done.

Line 532: The reference is missing the volume, issue, and page numbers.

Done.

Line 534: "Ssrn"appears to be incomplete journal name.

Done.

Line 539: The reference is missing the journal name.

Done.

Line 543: In the reference, capitalize the second "the".

Done. 

Reviewer #2:

As the title suggests that the manuscript presumably overviews the ability of different penalized regression methods in selecting predictive markers, it is not clear whether feature selection or predictability of selected features is focused. On the other hand, it is known that penalized regression methods are developed for variable selection instead of prediction, and there are a lot of theoretical results on the different methods (in contrast to the claimed "atheoretical predictive modelling approaches" by the authors). As long as the prediction is concerned, there are many machine learning methods (especially deep learning should be resorted). There are some other major issues:

We thank the reviewer for providing their thoughts on the current manuscript. We believe that we have presented a comparative examination of the current methods in terms of both prediction and feature selection, for which consideration in unison provides further insight into the differences between the methods. It is still desirable to have a subset of indicators which are accurate for prediction.

Penalised regression methods have, however, not been developed solely for the purpose of feature selection – for example, take ridge regression, which does not perform feature selection. The use of regularisation has benefits in terms of both prediction accuracy (related to the bias-variance trade-off) and model interpretability (e.g., feature selection). 

We do agree that there are a range of other machine learning methods (such as deep learning algorithms) that could also be used to develop predictive models; however, these models often have low interpretability (e.g., the black box problem) which can be important for many psychological applications (e.g., the development of screening tools).

1. The current overview solely relies on the analysis of single data set. Instead a comprehensive review may be better conducted with a well-designed simulation study, as well as several typical data sets. Otherwise, how have the authors calculated the sensitivity or specificity of selected predictors?

We have now acknowledged in the discussion that a simulation study is an important area of research that permits examination of sensitivity and specificity selected predictors (see amended text below). However, we believe there is great merit in comparing methods using real data, and this has been a focus of many studies including those comparing machine learning applications within the PLoS ONE journal (e.g., Luo et al., 2017; full reference provided below). The benefits of demonstrating using real datasets relate to illustrating how methods are applied in practice, for which the imperfections of real data and the accompanying decisions are present. As such, we believe our approach provides an important contribution to the literature and is consistent with the approaches taken by others to demonstrate methodology in practice. 

“First, the use of real data provides a relatable demonstration of how models may function, but findings may not necessarily generalise to other populations or data types. The use of simulation studies remains an important and complementary area of research for systematically exploring differences in predictive models (e.g., Pavlou et al., 2016).”

Luo, Y., Li, Z., Guo, H., Cao, H., Song, C., Guo, X., & Zhang, Y. (2017). Predicting congenital heart defects: A comparison of three data mining methods. PLoS ONE, 12(5), 1–14. https://doi.org/10.1371/journal.pone.0177811

2. The presented analysis of the longitudinal study is very vague, lack of details. For example, what is the response variable, how many predictors are there, and what is the sample size? It suggested in "Discussion" on "prediction of daily tobacco use in young adulthood", does it mean a binary response variable, or quantitative response variable? As the original study suggests a longitudinal data, is a cross-sectional data set selected for the analysis? Otherwise, how is the longitudinal data fit to the proposed models (again lack of details)?

We apologise if these details were unclear. We have now revised the wording throughout the methods section and believe that the design of the study is more clearly articulated. 

“Adolescent indicators

A total of 102 adolescent indicators, assessed at ages 13-14, 15-16, 17-18 years, by parent and self-report, were available for analysis (see S1 Table). Data spanned individual (i.e., biological, internalising/externalising, personality/temperament, social competence, and positive development), relational (i.e., peer and family relationships, and parenting practices), contextual (demographics, and school and work), and substance use specific (personal, and environmental use) domains. Repeated measures data were combined (i.e., maximum or mean level depending on the indicator type) to represent overall adolescent experience. All 102 adolescent indicators were entered as predictors into each model.

Young adult tobacco use outcome

Tobacco use was assessed at age 19-20 years (after measurement of the indicators), as the number of days used in the last month. This was converted to a binary variable representing daily use (i.e., ≥ 28 days in the last month), which was used as the outcome (response variable) in all analyses.”

3. Logistic regression is a regression model, instead of a statistical approach in parallel of penalized regression method. As logistic regression model can be used here, I suppose a binary response variable is used here. In this case, have the authors considered all penalized regression methods for the logistic regression models?

We have now clarified throughout that the penalised regression approaches were all penalised logistic regression models. We use standard logistic regression as a comparison to the penalised logistic regression models (i.e., the non-penalised comparison).

We haven’t considered all penalised regression methods, as mentioned in the limitations section. We do, however, believe that the models compared here are an inclusive selection of those used commonly throughout the literature, as reflected by their discussion in key texts (see James et al., 2013) and selection in other comparative studies (see Pavlou et al., 2016) . We have amended the text to read: 

“Three penalised regression methods which conduct automatic feature commonly compared in the literature and discussed in standard statistical texts (James et al., 2013; Pavlou et al., 2016) are the least absolute shrinkage and selection operator (LASSO) (Tibshirani, 1996), the adaptive LASSO (Zou, 2006) and the elastic-net (Zou & Hastie, 2005).”

---

## [Decision Letter · Decision Letter 1]

9 Nov 2020

A comparison of penalised regression methods for informing the selection of predictive markers

PONE-D-20-18792R1

Dear Dr. Christopher Greenwood,

We’re pleased to inform you that your manuscript has been judged scientifically suitable for publication and will be formally accepted for publication once it meets all outstanding technical requirements.

Kind regards,

Yuka Kotozaki

Academic Editor

PLOS ONE

Additional Editor Comments (optional):

Reviewers' comments:

Reviewer's Responses to Questions

**Comments to the Author**

1. If the authors have adequately addressed your comments raised in a previous round of review and you feel that this manuscript is now acceptable for publication, you may indicate that here to bypass the “Comments to the Author” section, enter your conflict of interest statement in the “Confidential to Editor” section, and submit your "Accept" recommendation.

Reviewer #1: All comments have been addressed

2. Is the manuscript technically sound, and do the data support the conclusions?

Reviewer #1: (No Response)

3. Has the statistical analysis been performed appropriately and rigorously? 

Reviewer #1: (No Response)

4. Have the authors made all data underlying the findings in their manuscript fully available?

Reviewer #1: (No Response)

5. Is the manuscript presented in an intelligible fashion and written in standard English?

Reviewer #1: (No Response)

6. Review Comments to the Author

Reviewer #1: (No Response)

7. PLOS authors have the option to publish the peer review history of their article (what does this mean?). If published, this will include your full peer review and any attached files.

Reviewer #1: No

---

## [Editor Report · Acceptance letter]

11 Nov 2020

PONE-D-20-18792R1 

A comparison of penalised regression methods for informing the selection of predictive markers 

Dear Dr. Greenwood:

I'm pleased to inform you that your manuscript has been deemed suitable for publication in PLOS ONE. Congratulations! Your manuscript is now with our production department. 

Kind regards, 

on behalf of

Dr. Yuka Kotozaki 

Academic Editor

PLOS ONE